# Land-use change interacts with climate to determine elevational species redistribution

Fengyi Guo [1], Jonathan Lenoir [2] & Timothy C. Bonebrake [1]

Climate change is driving global species redistribution with profound social and economic impacts. However, species movement is largely constrained by habitat availability and connectivity, of which the interaction effects with climate change remain largely unknown. Here we examine published data on 1464 elevational range shifts from 43 study sites to assess the confounding effect of land-use change on climate-driven species redistribution. We show that baseline forest cover and recent forest cover change are critical predictors in determining the magnitude of elevational range shifts. Forest loss positively interacts with baseline temperature conditions, such that forest loss in warmer regions tends to accelerate species' upslope movement. Consequently, not only climate but also habitat loss stressors and, importantly, their synergistic effects matter in forecasting species elevational redistribution, especially in the tropics where both stressors will increase the risk of net lowland biotic attrition.

[1] School of Biological Sciences, The University of Hong Kong, Pokfulam, Hong Kong SAR 999077, China. [2] UR "Ecologie et Dynamique des Systèmes Anthropisés" (EDYSAN, UMR 7058 CNRS-UPJV), Université de Picardie Jules Verne, 1 Rue des Louvels, 80037 Amiens Cedex 1, France. Correspondence and requests for materials should be addressed to T.C.B.tbone@hku.hk)

Human activities during the Anthropocene have trans-formed most of the planet, of which global forest loss and climate change are considered among the greatest threats to global biodiversity[1–5]. In response to these ongoing global changes, species are shifting their distributions to track suitable ecological niches along several geographic dimensions including latitude, longitude, and elevation/depth[6–8]. Due to the widely-recognized positive spatial autocorrelation signal of temperature conditions along latitudinal (it is warmer in the tropics and cooler towards the poles) and elevational/bathymetric (it is warmer at low elevation/depth and cooler at high elevation/depth) gradients, broad patterns of latitudinal and elevational/bathymetric range shifts are frequently linked to climate change effects[7,9–11]. For instance, Chen et al.[12] calculated the velocity of species range shifts as 16.9 kilometers per decade towards higher latitudes and 11 meters per decade towards higher elevations, presumably tracking temperature changes under global warming.

However, like temperature conditions, land use can also be positively autocorrelated in space[13,14], especially in mountain ecosystems where forest cover is not randomly distributed along elevational gradients (Fig. 1 and Supplementary Methods). The predominance of anthropogenic activities/disturbances in the lowlands[13] and the harsh climatic conditions prevailing at tree-line[15] constrain the proportion of forest cover per elevational band to peak at mid elevations (Fig. 1a). This pattern suggests that directional forest cover change (e.g., intensive deforestation in lowland areas common in Southeast Asia or forest expansion after land abandonment at high elevations in Europe) may be confounded with climate change when studying patterns of ele-vational range shifts for a large set of species[16]. Besides, small-scale species movements such as elevational range shifts could be largely constrained or confounded by local habitat availability[17–20]. Hitherto, very few empirical studies have looked into inter-acting effects between climate change and land-use change on the magnitude and direction of species range shifts[21–23] despite a wide recognition of potential synergistic and antagonistic effects[24–26].

Based on a recent and exhaustive review[7], we updated and extracted data from a set of 39 studies on climate-driven species range shifts to relate the rate of elevational range shifts against habitat and climate variables capturing baseline conditions, as well as temporal changes (Fig. 2). Making use of a high resolution global forest cover and forest change dataset[27], as well as the CHELSA climate dataset[28], we generated consistent and com-parable climate and land-use change indicators for each study to perform our analyses at two different resolution levels using either the data aggregated at the site level ($n = 43$; some of the 39 studies focused on several study sites that were treated inde-pendently here) or raw data at the species level ($n = 1464$). We found that apart from temperature changes and baseline tem-perature conditions, the rate of climate-related elevational range shifts is also affected by local habitat features such as baseline forest cover and recent forest cover change. The overall syner-gistic effects between climate and habitat change reveal the importance of considering multiple threats holistically when predicting biodiversity redistribution and for biodiversity conservation.

## Results

**Aggregated analysis.** Among the different linear models tested (see Methods) to explain the rate of species elevational range shifts averaged at the site level ($n = 43$), the model selection procedure yielded three candidate models of competing interest according to the corrected Akaike information criteria ($AIC_c$)[29] (cf. $\Delta AIC_c < 2$) (Table 1). All three ordinary least-square (OLS) regression models yielded similar $AIC_c$ values ($\Delta AIC_c < 0.2$) and Akaike weights (~0.3). One model in particular (Model 2 in Table 1), being the most parsimonious, had the simplest structure common to all candidate models explaining the majority of variation (OLS: $F_{4,38} = 5.76$, $R^2 = 0.31$, $P = 0.001$) (Table 2 and Supplementary Tables 1 and 2).

We found a consistent positive interaction effect (cf. synergistic effect) between forest loss and baseline temperature conditions across all three candidate models retained for the site-level analysis (Table 2), even after controlling for sampling effort in Model 2 (cf. sites weighted by the number of species included) (Supplementary Table 2). Under warmer baseline conditions and greater forest losses (e.g., in the tropics or lowland areas), species

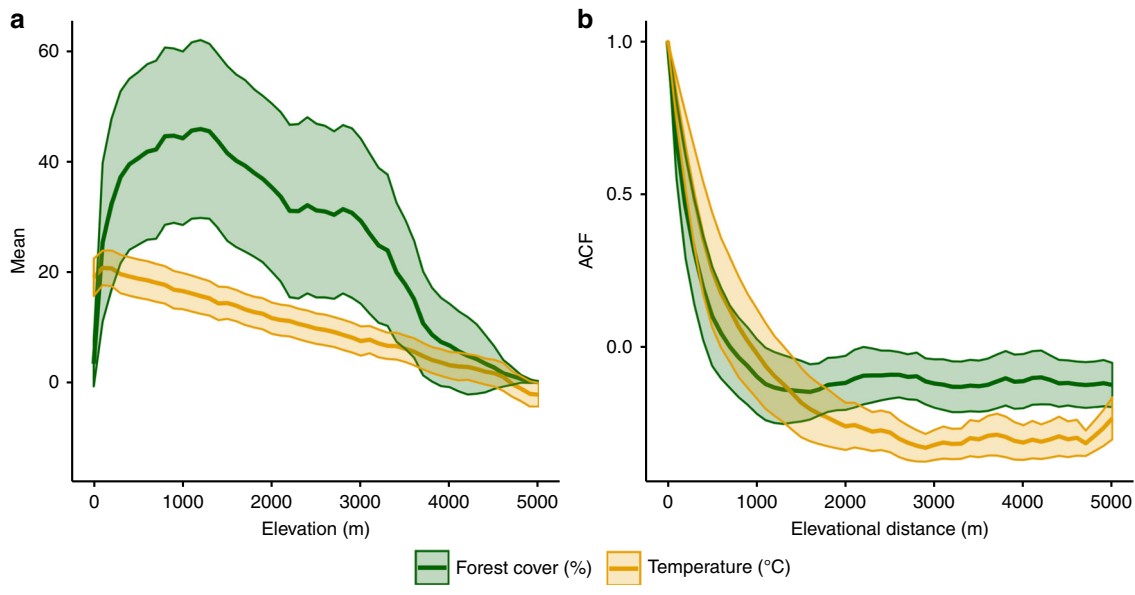

**Fig. 1** Spatial autocorrelation signal of forest cover and temperature across elevation. **a** General pattern of forest cover (%) and temperature (°C) per 100 m elevational band for 140 global mountain ranges. **b** The associated autocorrelation function (ACF) displaying the elevational autocorrelation signal of the proportion of forest cover and temperature along the elevational gradient. Lines and shade each represent the mean and ± 0.5 S.D. boundaries. For details on calculation refer to Supplementary Methods

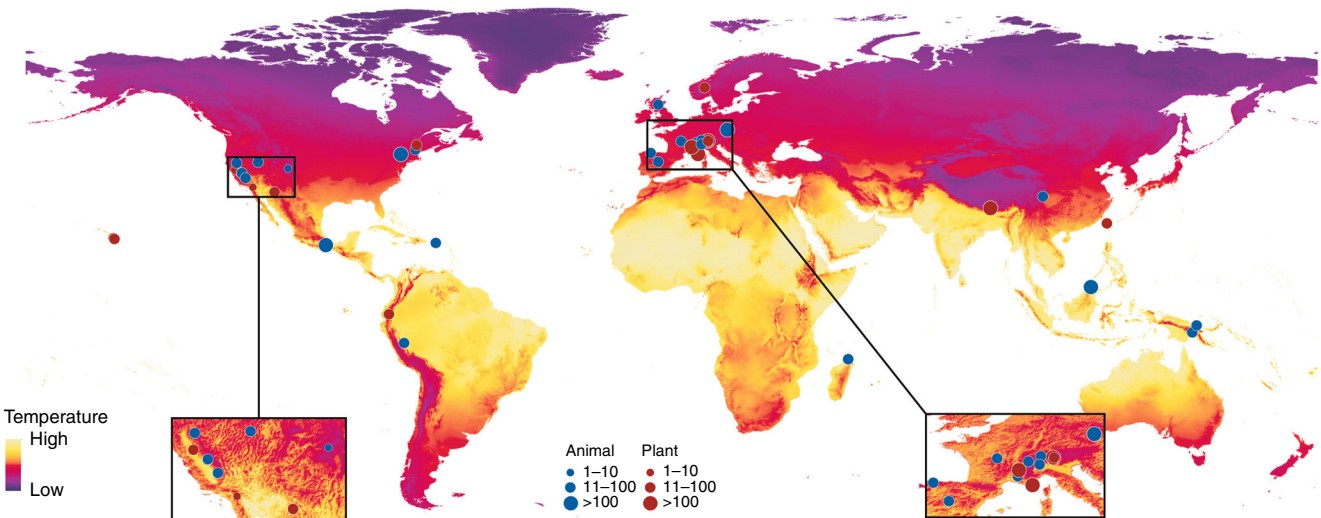

**Fig. 2** Locations of the 43 distinct study sites on climate-related elevation shifts. Base map created using mean temperature data from CHELSA[28]. Sub-frame shows the zoomed-in details of study sites in North America and Europe. Blue dots represent studies on animals and red dots studies on plants. Dot size varies by number of species resurveyed. Data are collected from 39 selected studies. For details of each study refer to the Supplementary References

<table>
<tr><td colspan="5"><b>Table 1 Candidate linear models on the site average shift rate</b></td></tr>
</table>

| Model | Variables | AICc | ΔAIC$_c$ | Weight | $R^2$ |
|---|---|---|---|---|---|
| 1 | Loss, Cover, T, Loss × T, Loss × Cover | 357.2 | | 0.35 | 0.34 |
| 2 | Loss, Cover, T, Loss × T | 357.3 | 0.08 | 0.33 | 0.31 |
| 3 | Loss, Cover, T, Loss × T, Type, Loss × Type | 357.3 | 0.16 | 0.32 | 0.36 |

Candidate linear models with interactive effects between climate and habitat features on the site average shift rate ($n = 43$), ranked by the corrected Akaike information criteria (AICc)
T: baseline temperature, Loss: forest loss percentage, Cover: forest cover percentage, Type: taxa type (animal or plant)

**Table 2 Model details for the site average shift rate**

| Parameter | Estimate | Std. error | t value | Pr (>|t|) |
|---|---|---|---|---|
| *Model 1* | | | | |
| Intercept | 11.24 | 2.46 | 4.58 | <0.001 |
| scale (Loss) | −5.80 | 2.93 | −1.98 | 0.06 |
| scale (Cover) | 6.85 | 2.69 | 2.54 | 0.02 |
| scale (T) | 12.94 | 3.24 | 3.99 | <0.001 |
| scale (Loss) × scale (T) | 15.98 | 4.52 | 3.53 | 0.001 |
| scale (Loss) × scale (Cover) | 5.84 | 3.61 | 1.62 | 0.11 |
| *Model 2\** | | | | |
| Intercept | 11.09 | 2.51 | 4.43 | <0.001 |
| scale (Loss) | −7.78 | 2.71 | −2.87 | 0.007 |
| scale (Cover) | 5.87 | 2.68 | 2.19 | 0.03 |
| scale (T) | 11.61 | 3.20 | 3.63 | <0.001 |
| scale (Loss) × scale (T) | 15.79 | 4.62 | 3.42 | 0.002 |
| *Model 3* | | | | |
| Intercept | 8.68 | 2.97 | 2.92 | 0.006 |
| scale (Loss) | −3.36 | 3.25 | −1.04 | 0.31 |
| scale (Cover) | 6.99 | 2.64 | 2.65 | 0.01 |
| scale (T) | 13.60 | 3.21 | 4.24 | <0.001 |
| Type_plant | 2.88 | 4.62 | 0.62 | 0.54 |
| scale (Loss) × scale (T) | 18.16 | 4.76 | 3.82 | <0.001 |
| scale (Loss) × Type_plant | −9.75 | 4.44 | −2.20 | 0.03 |

Details of the three best fitting models (Table 1) for the site average shift rate ($n = 43$). Predictor variables are scaled (cf. the scale() function in R) for comparison purposes
\*See Supplementary Tables 1 and 2 for unscaled estimates and weighted coefficients of model 2, the most parsimonious model
T: baseline temperature, Loss: forest loss percentage, Cover: forest cover percentage, Type: taxa type (animal or plant)

tend to shift more rapidly towards higher elevations while under colder baseline conditions (e.g., in boreal regions or high elevation zones), the effect of forest loss is reversed: species tend to shift less rapidly along the elevational gradient and even towards lower elevations (cf. negative shift rates) (Fig. 3).

We found additional, but not consistent interactions in the other two OLS regression models (Table 1). The positive interaction between forest loss and baseline forest cover (Model 1) was not significant itself, although the overall $R^2$ was slightly improved (Table 2). The interaction term between forest loss and taxa type (Model 3) was marginally significant (Table 2) with elevational range shifts being lower in magnitude for plants than for animals under high forest loss (Supplementary Fig. 1). However, this weak trend was likely driven by a single data point attributed to changes in water balance[30] (Supplementary Fig. 1).

**Disaggregated analysis**. Among the different mixed-effect models we tested (see the Methods section) to explain the rate of species elevational range shifts at the species level ($n = 1464$), the model selection procedure yielded 30 candidate models of competing interest (cf. ΔAIC$_c$ < 2) (Supplementary Table 3). The generally similar Akaike weights across the 30 competing models suggest that we cannot select one single "best" model. Hence, and because of the large number of competing models, we here rely on the model-averaged coefficients based on the 30 selected models to assess the effect of different predictors on elevational shift rate at the species level (Fig. 4). Our findings at the species

level suggest increasing shift rates for species with higher baseline temperature conditions, and for greater elevational distance to the highest mountain summit within the study area. Although the data suggest that the magnitude of the elevational shift rate might be higher under denser baseline forest cover conditions, this effect was not significant (see the 95% confidence intervals crossing the zero line for "Cover" in Fig. 4). The data also suggest that the magnitude of the elevational shift rate might be affected by synergistic effects between climate change rate and baseline temperature, but these effects were also not significant (see the 95% confidence intervals crossing the zero line for "CCR" and

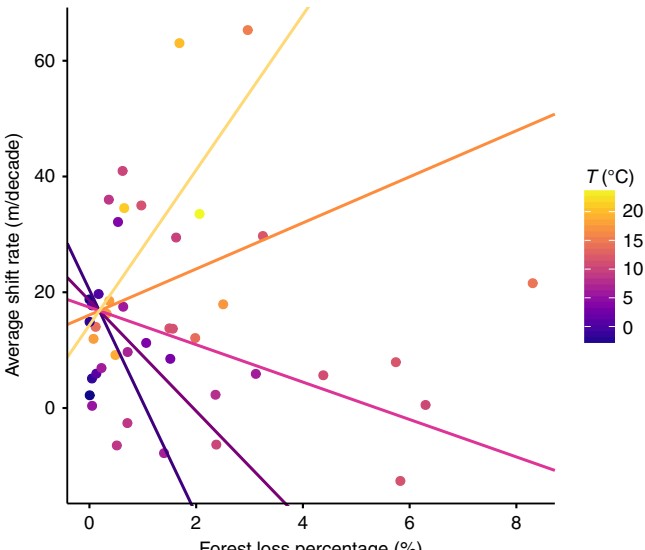

**Fig. 3** Synergistic effect between forest loss and baseline temperature on species elevational shift rate averaged at the site level. Data from $n =$ 43 sites were plotted in a natural scale for a straightforward display of the relationship between each explanatory variable and the average shift rate. Each of the five colored regression lines represents different baseline conditions in temperature ($T$). Note that other covariates in this model (cf. Model 2 in Tables 1 and 2) were set to their mean values

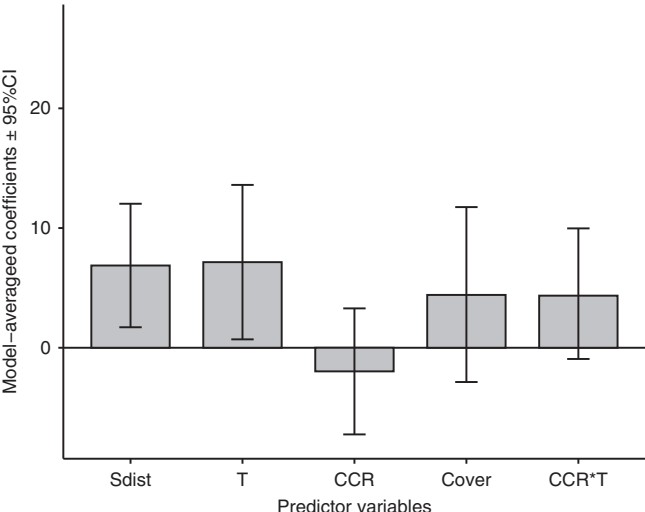

**Fig. 4** Coefficient averages of the five most important predictors with 95% confidence intervals. Models constructed using the full dataset (n = 1464). For details of the 30 competing models refer to Supplementary Table 3. All variables are scaled to allow direct comparison both in direction and in magnitude, ranked by importance (threshold = 0.6). Sdist: distance to mountain summit, T: baseline temperature, CCR: climate change rate, Cover: forest cover percentage

"CCR * T" in Fig. 4). Interestingly, when restricting data to forest systems only (forest cover >25%, $n = 1120$), we found significant synergistic effects between climate change rate and forest loss percentage on elevational shift rate at the species level (Supplementary Fig. 2). Therefore, reflecting our findings at the site level, both climate and land-use change factors are important in driving species upslope movement within forest systems.

## Discussion

We found evidence for the confounding impacts of habitat features (e.g., forest cover, forest loss) on climate-related species range shifts across elevation using both an aggregated (site level) and a disaggregated (species level) approach. At the site level, we found a robust and consistent synergistic effect between baseline temperature conditions and forest loss across all models. Although forest loss in colder regions of the world is likely to impede species upslope movements, potentially by interrupting habitat connectivity[19], this negative effect is somewhat reversed in warmer regions where high forest loss tends to increase the elevational shift rate towards higher elevations. The mechanisms behind this synergistic effect of lowland deforestation on climate-driven upslope range shifts in warm environments are not yet known but may involve direct population extirpation of plant species, drastic microclimatic changes (e.g., increasing temperature and decreasing humidity) following habitat disturbance[21,26,31,32] and, ultimately, local extinction at the lowest margin of species' elevational ranges[33]. This result supports suggestions of rapid upslope range shifts through lowland biotic attrition[34], especially in tropical lowland areas experiencing both warm temperature conditions and the most severe contemporary human disturbance impacts in terms of deforestation[13,27].

In both analyses (i.e., aggregated and disaggregated), the generally positive effects of forest cover (although not significant at the species level) on the elevational shift rate reveal the importance of habitat cohesion and availability for species' movement under climate change impacts[18,20,24]. The fact that warm-adapted species tend to shift more rapidly upslope compared to those in

cooler regions could be explained by seasonal temperature stability, i.e., they need to move greater elevational distances to reach the suitable and very stable environment to which they are narrowly adapted to and which corresponds with their restricted thermal niches[35,36]. Alternatively, it could also be partially driven by the shallow temperature gradient across latitude in the tropics, which makes latitudinal shifts less likely than elevational shifts towards mountain summits[34]. The rate of climate change over time is not a major predictor in most models, likely due to the low spatial resolution of climate data, especially for mountainous regions. However, we did observe a positive interaction between forest loss percentage and climate change rate for the disaggregated analysis, when restricted to forest systems. Overall, the above-mentioned interactions emphasize the fact that isolating the effects of co-occurring stressors and ignoring their potential synergistic or antagonistic effects could be misleading[21,25].

The elevational distance to the highest mountain summit within the study area was also found to be an important factor in explaining upslope shift rates at the species level: the greater the elevational distance to the highest summit, the more rapidly species shifted their range upslope. This finding supports the idea that limited physical space may constrain the rate of species' range shifts along elevational gradients[16,37,38]. In addition, microclimate conditions and soil nutrition levels determined by montane mass (the Massenerhebung effect) could also limit species range extension[39,40].

One limitation to our work lies in the short time-span of the forest data used (2000–2015), especially given that most selected studies covered periods of several decades. On average, initial surveys were conducted 44 years prior to the period covered by the Global Forest Watch database. Land-use history over the last decade is not likely to accurately mirror changes over previous decades, and this mismatch in time scale might have contributed to the unexplained variance in our models at the species level (cf. 87.4% on average, with ±0.97% S.D.). Nonetheless, the spatial consistency that the Global Forest Watch dataset offers is a breakthrough for large-scale comparisons in habitat features (see discussion on model limitations using various habitat data

sources in Mantyka-Pringle et al.[25]), and the baseline forest cover data in year 2000 might also represent previous land-use histories that were missed in terms of forest change information. Apart from the forest cover dataset, the elevational shift data analyzed here are also likely biased towards species that have exhibited observable responses to climate change. Other factors such as precipitation change, individual species' physiology and biotic interactions may also greatly impact shift patterns[30,41–44]. Geographic bias, especially the relative lack of tropical studies, is another limitation here, which also characterizes former findings[7]. Moreover, the fact that most tropical studies in our dataset were from islands, whereas temperate studies were largely continental could bring in additional bias in terms of biotic composition. However, because our aim was to investigate the overlooked interaction between climate and land-use change on species redistribution, instead of drawing a conclusive quantitative model, we believe the data used here are suitable.

To conclude, we found large and consistent impacts of habitat features, indicated by forest cover and forest cover changes, on the rate of species' elevational range shifts. Furthermore, we also discovered significant interactions between land-use change and climate variables in driving species' upslope movement. Therefore, species elevational redistributions cannot be attributed to warming impacts alone, although climate change is an important factor. Assessments must consider not only climate (both the baseline conditions and the magnitude of the change) but also habitat loss stressors and baseline conditions and, importantly, their synergistic effects in conservation planning and management.

## Methods

**Data collection.** We selected 39 studies on climate-related elevational range shifts (Fig. 2) as synthesized by Lenoir and Svenning[7] and by searching literature published between 2014 and 2017 following the same protocol as in Lenoir and Svenning[7] (see Appendix 1 therein). Data were only included if observed elevational range shifts were reported in the original research. When more than one geographically distinctive site was surveyed in a study with shift information separately reported (e.g., Freeman and Freeman[45]), we treated them as different sites ($n = 43$). Sites with fewer than 5 species resurveyed were excluded for the sake of representativeness. In addition to some general information (e.g., location, kingdom/taxon, species number, starting and ending year, study period calculated by subtracting start year from end year) at the study level, which is tabulated in Lenoir and Svenning[7] (Table A3), we also recorded the elevation range covered by each focal study, together with detailed species' elevational range shifts as reported in the original focal study (positive values for upslope movements and negative for downslope shifts). When only significant shifts were reported, we calculated the average shifts based on the ratio between shifting species and the total number surveyed. When raw data were not available, we used the "DataThief" program (http://datathief.org/) to extract the raw data from figures. In terms of different reference points (e.g., upper limit, lower limit or midpoint), we used a simplified binary classification (center vs. margin); when possible, shift data on optimum position were preferred, as they are less sensitive to sampling effort[46] and statistically more robust compared with shift data at the distribution margins[47]. Data on marginal reference points were included if there was no better alternative (order of preference: optimum/mean/midpoint > margins).

As indicators of land-use change for each mapped location[5], we extracted high resolution (30 m resolution at the equator) baseline forest cover (year 2000), forest cover loss (2000–2015) and forest cover gain (2000–2012) data from Global Forest Watch[27] (http://earthenginepartners.appspot.com/science-2013-global-forest/download_v1.3.html). To assess habitat alteration and species distribution shift patterns for specific elevation ranges, we overlaid these forest layers onto fine scale digital elevation models (SRTM Arc-Second Global, 30 m resolution at the equator, https://lta.cr.usgs.gov/SRTM1Arc) for each site (in one Norway site with no SRTM data, a local digital elevation model with 10 m resolution was obtained from Norwegian Mapping Authority, Kartverket, https://www.kartverket.no/en/data/Open-and-Free-geospatial-data-from-Norway/) and exported forest data at intervals of 10-m elevation bands (See Supplementary Fig. 3 for an example). Therefore, baseline forest cover percentage and forest cover change profiles by 10-m elevation bands along the elevational gradient could be generated for each study site for subsequent analyses (Supplementary Figs. 4 and 5).

Climate data were obtained from CHELSA Version1.2[28] and CRU_TS4.00[48]. We used the annual mean temperature from 1979 to 2013 (cf. bio1 at 30 arc-second which is about 1 km resolution at the equator, http://chelsa-climate.org/downloads/) as the baseline temperature for each site. We also computed

temperature data for each 10-m elevation band by overlaying it onto digital elevation models using the same method as we did for the forest data. Time-series data on temperature from CRU (0.5°, ~55 km at the equator, https://crudata.uea.ac.uk/cru/data/hrg/cru_ts_4.00/ge/) were analyzed to estimate site-specific climate change rate (°C per decade) by fitting the regression line of temperature (response variable) against time (predictor variable) over each study period, following Chen et al.[12]. When the study area was too large for fine grid calculation, and considering that site-specific temperature data were obtained by averaging across all grid cells covering the study area (which were less sensitive to resolution differences), we used the 5° resolution temperature anomaly data which is ~550 km at the equator (CRUTEM4[49,50], https://crudata.uea.ac.uk/cru/data/crutem/ge/) to run the regression over time. Both climate and habitat data were obtained using Google Earth and ArcGIS 10.2.

**Aggregated analysis.** For each of the 43 sites (from 39 selected studies), the average elevational range shift (m) for all examined species across the spatial extent of the focal site was either reported as in the original paper or calculated from published raw data. To account for the fact that each study focused on a different period (cf. starting and ending years were study-specific), we calculated the average elevational range shift rate (m per decade) as the response variable for cross-site comparisons. We ran ordinary least-square (OLS) regressions to assess the potential confounding effect of land-use change on "climate-driven" elevational range shifts. As explanatory variables, we used site-specific climate data (T: baseline temperature based on annual mean conditions between 1979 and 2013 (°C), CCR: climate change rate (°C per decade)), average forest data within the elevation range of the focal study (Cover: baseline forest cover (%), Loss: forest cover loss (%), Gain: forest cover gain (%)), as well as other recorded parameters, including taxon (plants vs. animals) and reference point (center vs. margin). All numeric variables were scaled for effect size comparisons[25,51]. Prior to analysis, we checked for data collinearity by calculating the variance inflation factor (VIF) and applied a cutoff threshold of 2[52]. No signal of covariate collinearity was detected with this pre-selected, conservative threshold (see Supplementary Table 4 for correlation matrix). We used the "stepAIC()" function in the R package MASS[53] to select (in both directions: forward and backward) for the models with the smallest Akaike information criteria (AIC) values, with interactions specified up to two degrees between variables. Given the fact that stepwise model selection process does not necessarily generate the most meaningful model, especially for limited sample sizes[29,54], we manually added/removed parameters and interaction terms that appeared in the last round of model selection to/from the top model to generate candidate models for comparison. The best of these candidate models was determined based on $AIC_c$ (corrected for finite sample size), parsimony and adjusted $R^2$ values. We inspected diagnostic residual plots to assess goodness of fit of the best model (Supplementary Fig. 6). All analyses were performed in RStudio[55].

**Disaggregated analysis.** Species within each mountain range were found to exhibit very diverse range shifts, both in magnitude and direction[41], likely associated with their elevational locations. Therefore, we retrieved all species-level shift data ($n = 2798$) to construct disaggregated models of species range shifts, with individual species' shift rates as the response variable. The species-level shift dataset is dominated by records from a single study site (Dainese et al.[23], $n = 1334$ out of 2798), of which the rapid shift rates were associated with proximity to roads and the occurrence of several non-native species. The goodness of fit of models including these data points are low ($R^2_{conditional} < 0.02$, $R^2_{marginal} < 0.05$). Therefore, we excluded data from Dainese et al.[23] and only used data from other study sites in the following analyses ($n = 1464$). We located different species onto specific elevation bands (10-m interval) based on their historical central/marginal location along the elevation gradient, as reported from the original study. We then matched the species-level shift rate with temperature and habitat features (baseline forest cover and forest cover change) with reference to their elevational locations (See Supplementary Fig. 3 for an example). As the raw mean shift extent across species was 66 m, we constructed a 100-m buffer for both forest and temperature data by calculating the area-weighted mean of five bands above and below each 10-m elevation band, and used the buffered mean as the overall representation of surrounding microclimate features. When temperature data were missing for certain elevation bands (due to the coarser resolution compared with forest data), we estimated temperature values by calculating the mean local elevational lapse rate, based on the temperature at other elevations of the same site. Climate change rate, however, was a more regional feature and thus remained the same as in the aggregated analysis (assuming all species located in the same study site experience the same climate change rate).

Applying the same VIF threshold to check for data collinearity, the forest gain variable (with VIF > 2) was excluded in the following analyses (see Supplementary Table 5 for the correlation matrix) due to its positive correlation with forest loss. According to the Global Forest Watch database, forest gain is defined as a change from a non-forest to forest state, and unlike the annually allocated forest loss information, forest gain percentage was calculated over the entire study period (2000–2012) due to the slow growing process[27]. Natural growth of existing long-lived forest was not considered as forest gain[27]. Therefore, forest loss and gain are often correlated in areas with temporary deforestation (e.g., logging, harvesting, fire, etc.) followed by replantation or natural regrowth. Based on this, we decided to

focus on forest loss solely as it better reflects the dynamic of habitat disturbances. To account for the nested design of our disaggregated dataset (cf. several range shift values for a given single study site) and the potential pseudo-replication issue of having identical values for the set of variables available only at the study site level (e.g., climate change rate), we ran linear mixed-effect models (LMMs) with "Site" as a random factor potentially affecting the intercept (i.e., 1|Site) and the slope parameters of the elevation band-specific predictor variables (i.e., 1 + Cover|Site, 1 + Loss|Site, 1 + T|Site, as well as additive combinations including 1 + T + Cover| Site, 1 + T + Loss|Site and 1 + Cover + Loss|Site). We used the "lmer()" function from the lme4 package in R[56] to run LMMs. The optimal random component structure among the previously mentioned structures was determined by comparing $AIC_c$ values of LMMs fitted by restricted maximum likelihood (REML), while the fixed effects were kept constant with all possible explanatory variables and two-way interaction terms incorporated (cf. the "beyond optimal model" sensu Zuur et al.[51]). Using REML is generally considered to give better estimates for the random effects. Candidate fixed effect variables were similar to the aggregated approach (all standardized), although most data were elevation-band-specific, whereas climate change rates were still averaged across the whole study. We also calculated the elevational distance of each species' location to the highest mountain summit (Sdist) available within the study area as an additional explanatory variable in the fixed effect terms. The reasoning behind this variable is that a species' elevational optimum/margin that is already located very close (cf. in terms of vertical distance) to the highest mountain summit available within the study area is more constrained to shift upward than a species for which the elevational optimum/margin is located further away. The optimal random effect structure we found when running all seven beyond optimal models mentioned above suggests that forest cover tends to have site-specific impacts on individual species' shift rates (i.e., the 1 + Cover|Site random structure showed the lowest $AIC_c$ value). Based on this optimal random effect structure, the fixed effect component was then modified using the "dredge()" function available from the R package MuMIn[57] for model selection. To compare models with nested fixed effects (but with the same random structure: here 1 + T + Cover|Site), we used maximum likelihood (ML) estimation instead of REML, as recommended by Zuur et al.[51]. Based on the outcomes of the "dredge" function, the relative variable importance was calculated as the sum of Akaike weights across selected models, using the "importance()" function and after refitting all selected models using REML for final inference and reporting of the models' parameters[51]. We also calculated the model averaged coefficients and their bootstrapped confidence intervals to identify significant relationships. We inspected diagnostic residual plots to assess goodness of fit of the top model (Supplementary Fig. 7). All analyses were performed in RStudio[55].

**Sensitivity analysis restricted to forest systems only**. To test whether the forest change data we used could adequately represent elevational habitat changes in general, we reran the above analyses at both site ($n = 29$) and species ($n = 1120$) levels on forest systems only, by restricting data to locations with average forest cover greater than 25%, and to sites with no fewer than 5 species resurveyed. Results were largely consistent with those across the entire forest cover gradient (Supplementary Table 6 and Supplementary Fig. 2).

**Data availability**. Supplementary Data 1 is the data used for site-level analysis, and Supplementary Data 2 for species-level analysis. Both datasets are available on Dryad [https://doi.org/10.5061/dryad.k8g2672].

**Code availability**. R script for running the analyses are available as Supplementary Software.

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

## Acknowledgements

This work was supported by the Research Grants Council General Research Fund (HKU 760213).

## Author contributions

T.C.B. and F.G. conceived the project; F.G. complied the database with input from J.L., F. G., T.C.B., and J.L. analyzed the data and F.G. wrote the first draft of the paper with substantial contributions from T.C.B. and J.L.

## Additional information

**Competing interests:** The authors declare no competing interests.

