## [Peer Review File(PDF 363 kb) · Nature Communications]

Reviewers' comments:

Reviewer #1 (Remarks to the Author):

I was excited to read the paper by Guo and colleagues as I also work on climate and land-use change interactions on biodiversity and agree that very few empirical studies have looked into interacting effects between climate change and land-use change on the magnitude and direction of species range shifts. I commend the authors on tackling such a thoughtful paper, but I have a few questions and suggestions for the authors which are important for the modelling and interpretation of the results:

1) The study size is limited by sample size – only 29 studies included and only 18 of these are determined actual forest ecosystems (>25% forest cover). Not all studies are from mountainous country (e.g. Breeding bird species in NY; Zuzkerberg et al. (2009)) so the title of the paper should be rethought – might be more appropriate to say elevational range shifts or elevational species redistribution.

2) Were studies weighted by the number of species included? E.g. Zuckerberg et al. (2009) included 129 bird species in their analysis whereas Rowe et al. (2009) only studied 25 small mammal species. If no, why not?

3) Why weren't the other models reported in Table 1 of the supplementary material? This is essential for understanding how much better the candidate models explain the data relative to other models.

4) The relationship between CCR and average shift rate was not tested in the sensitivity analysis with only the forest ecosystem sites (n = 18) due to small sample size. The authors therefore really need to downplay this interaction or reanalyze with a larger dataset as it is currently not convincing. Note that forest loss and forest cover coefficients were also no longer positive with the +/- 95% CI (Supp. Fig. 4). This should be mentioned in the 'Sensitivity Analysis' section of the paper.

5) Why not display the model-averaged coefficients for different taxa type (animal or plant) since 'type' was a parameter in two of the competing models? It would also be valuable to split grassland versus forest dependent species (and maybe tropical) as I suspect that they responded differently to forest cover (%). Forest loss may also be correlated with other variables such as a gain in grassland cover and/or wetlands which could explain higher/faster movements for grassland and wetland dependent species. This additional piece would also probably help to explain the unexplained variance in the models at the species level.

Minor comments:

Elsen & Tingley (2015) reference is missing from supplementary material.

Pg 5, Line 84 – the variables of the best fitted model should be stated here.

Reviewer #2 (Remarks to the Author):

The authors looked at confounding effects of forest cover change on climate-driven range shifts across elevational gradients by compiling and re-analyzing published data. I think it is a much needed work even though studies focusing on climate-driven redistribution should have tried their best to exclude confounding factors, particularly habitat change, in the original design. In reality, it is challenging and not feasible for many studies. Thus, evaluating the interacting effects is a very nice attempt and provides insights into the widely reported biological responses under climate change.

After reading through the manuscript carefully, I think, generally, the authors analyzed the data appropriately given the question to address and data available so far. The discussion and conclusion followed logically and coherently. They also made it clear about the limitation of the study. However, I am not yet fully convinced by the interpretation of the results because of the constraints of multiple regressions, given the small sample size and some data treatment issues.

The authors have probably included most data relevant for addressing the research question. However, comparing to the number of predictors, the sample size was nevertheless small and the results may be sensitive to the data. There may exist multicollinearity or indirect effects among predictors. For example, higher forest cover (represented by "Cover" in the model) toward the tropics (baseline temperature, "T"), or, as the authors suggested in line 45-47, stronger deforestation ("Loss" in the model) toward the tropics. Indirect effects may occur for average temperature affects the level of forest cover and then buffers species responses. Both of them influence the robustness of multiple regressions and the indirect effect cannot be captured by current analysis setting. The authors could easily check the collinearities of variables. As for the indirect effect, however, the current sample size may still impede meaningful analysis.

The authors also conducted disaggregated analysis, which tried to look at species-specific responses and also increased sample size. However, because of the coarse spatial resolution of the climate data, a single temperature, identical to the baseline temperature for aggregated analysis, was applied to each species in the site of interest. The constraint made it difficult to test the authors' idea that lowland deforestation and/or forest expansion at highland may interact with climate to affect range shifts. The mismatch between species level response and their corresponding temperature also made the interpretation of baseline temperature ambiguous. The elevational gradients could easily represent comparable temperature ranges along latitudinal gradients. Assigning an average temperature dramatically reduced the variation to site difference. It was thus not surprising that the disaggregated analysis derived generally similar results from aggregated analysis and unfortunately added limited information.

I had a closer look at the raw data and would like to know more details about the data treatment. As far as I understand, many studies looked at higher part of the elevational gradients to avoid impacts of lowland deforestation. For example, Jump et al. (2012) looked at plants at summit areas. Raxworthy et al. (2008) analyzed species distributed higher than about 1400 m.a.s.l. However, in Supplementary Figure 3, the authors seemed to include lower part of the gradient that was outside the focal extent of the original study. I did not check all of them. Please make it clear whether all the forest change profiles corresponded to the study gradients. This is where the authors argued not being considered in the previous studies but they might have done.

For the aggregated analysis, Chen et al. (2009) and Chen et al. (2011) both looked at moth assemblages in Mount Kinabalu and shouldn't be included twice. For disaggregated analysis, the number of species of each study included in the final analysis differed a lot from the aggregated analysis. The authors included 21 data points of Chen et al. (2011) from the original 130 species, 22 data points of Jump et al. (2012) from the original 24 species, 15 data points of Konvicka et al. (2003) from the original 119 species, 21 data points of Parolo and Rossi (2008) from the original 93 species...etc. For Walther et al. (2005), more species than originally reported were included (24 rather than 18). Please clarify if there were criteria to include species.

More than half of the studies focused on plants. As the authors suggested, forest change implied habitat alteration and decreasing buffering effect on animal species under warming. However, for the plants, the gain and loss of forest may reflect direct population extirpation or increase. Shall we give broadly similar explanations to the underlying mechanisms?

Reviewer #3 (Remarks to the Author):

Guo and colleagues follow up on the landmark global study of Lenoir and Svenning (2015), who analysed 1030 elevational range shifts from 29 published studies. Here, the focus is on the impact of baseline forest cover and recent forest cover change (which they assessed using the Global Forest Watch database for post-2000 changes) on these documented range shifts, and on the interaction between these factors and baseline temperature and temperature change (estimated specifically for each study). They analysed the data in two ways—at the study level, and at the species level (while controlling for nested nature of the data).

I should say, first, that reading the Methods (when I finally got them) reassured me that authors had successfully steered clear the many statistical pitfalls that had come to mind as I read the Introduction and the Results. The analysis appears quite sound, and the statistical inferences (given the data—see below) appropriately conservative.

The authors found clear statistical evidence of a synergistic (amplifying) interaction between the forest factors and the climate factors. Few will be at all surprised by this finding, as it has been frequently predicted, but as far as I know this study is the first to offer rigorous evidence.

Less predictable (and thus potentially more important) is their finding that “forest loss positively interacts with baseline temperature conditions such that forest loss in warmer regions tends to accelerate species upslope movements.” Unless I am mistaken, this conclusion relies rather heavily on the only three tropical sites among the 29 sites: Hawaii (Volcanoes NP), Borneo (Mt. Kinabalu), and Madagascar. This modest site-level sample size is no fault of the authors—as far as I know, those three studies are the only tropical ones so far available that meet the Lenoir and Svenning’s (2015) selection criteria. The authors mention the small number of tropical sites as a limitation of their study (Line 186 ff), but I confess that their reassurance on this point left me scratching my head: “However, because our aim was to investigate the overlooked interaction between climate change and land-use change on species redistribution, instead of drawing a conclusive numerical model, we believe the data used here are suitable.”

More troubling, perhaps, is that all three tropical sites are large islands, each famous in its own right for endemism and notorious for its lowland habitat destruction. How this fact might affect the results is not clear, but the potential bias should at least be discussed. For Hawaii and Madagascar (more isolated than Borneo), highland biotas (if memory serves) are believed to have been derived from lowland immigrants, which is surely not the case with most continental sites. (There is an excellent paper on the origins of the Mt. Kinabalu biota, but I can’t seem to put my hands on it.)

The authors note that “greater elevational distance to the highest mountain summit within the study areas” (Line 168) had an effect on the results, but they do not mention the Massenerhebung effect (see Grubb 1971, McCain 2005), which may well be relevant to this finding. (Mt. Kinabalu is, in fact, a classic case of this effect.)

Connectivity is (appropriately) mentioned as a critical issue in the Abstract, but not adequately addressed in the body of the paper. Does the Global Forest Watch data somehow take connectivity into account? Assessing connectivity quantitatively for each study area (the novel method of Ferrero-Medina et al. 2010, cited by authors, might potentially be applied) may be too much to ask for this study, but it should be discussed.

Some smaller things:

Fig. 3 a and b. I like the clever way of plotting interactions, but the caption does not make clear that the two panels plot the results for different models, and why they differ. It took me way too long to figure it out from the text.

Line 158. "The fact that warm-adapted tropical species tend to shift more rapidly upslope compared to those in cooler regions could be explained by seasonal temperature stability, i.e. 'mountain passes are higher in the tropics' (Janzen 1967; Bonebrake 2013)." I am baffled by this suggestion. Janzen's point was precisely the opposite: in the tropics, elevational shifts (e.g. going up an over a mountain pass) are more challenging than at higher latitudes because species are more narrowly adapted to local (stable) temperature limits.

Line 161. "Alternatively, it could also be partially driven by the shallow temperature gradient across latitudes in the tropics, which makes the other option, elevational shift towards mountain summits, more likely...." It might be more accurate to say, "Alternatively, it could also be partially driven by the shallow temperature gradient across latitudes in the tropics, which makes latitudinal shifts less likely than the other option, elevational shift towards mountain summits."

Grubb, P. 1971. Interpretation of the 'Massenerhebung'effect on tropical mountains. *Nature* 229:44-45.

McCain, C. 2005. Elevational gradients in diversity of small mammals. *Ecology* 86:366-372.

Robert K. Colwell
University of Connecticut

Reviewers' comments:

Reviewer #1 (Remarks to the Author):

I was excited to read the paper by Guo and colleagues as I also work on climate and land-use change interactions on biodiversity and agree that very few empirical studies have looked into interacting effects between climate change and land-use change on the magnitude and direction of species range shifts. I commend the authors on tackling such a thoughtful paper, but I have a few questions and suggestions for the authors which are important for the modelling and interpretation of the results:

1) The study size is limited by sample size – only 29 studies included and only 18 of these are determined actual forest ecosystems (>25% forest cover). Not all studies are from mountainous country (e.g. Breeding bird species in NY; Zuzkerberg et al. (2009)) so the title of the paper should be rethought – might be more appropriate to say elevational range shifts or elevational species redistribution.

FG et al: Thanks for the suggestion, we changed the title from “montane species redistribution” to “elevational species redistribution”. And the sample size has substantially increased to 43 sites with 29 identified as forest systems (major change #1).

2) Were studies weighted by the number of species included? E.g. Zuckerberg et al. (2009) included 129 bird species in their analysis whereas Rowe et al. (2009) only studied 25 small mammal species. If no, why not?

FG: We have run additional analysis for the site-level data with sites weighted by number of species included and the results are consistent (see the new Supplementary Table 2). Furthermore, we believe that the disaggregated (species-level) analysis can well-capture the effects of different sample sizes among studies.

3) Why weren't the other models reported in Table 1 of the supplementary material? This is essential for understanding how much better the candidate models explain the data relative to other models.

FG: We appreciate the suggestion and have included the details of each of the candidate models in the main text (see the new version of Table 2).

4) The relationship between CCR and average shift rate was not tested in the sensitivity analysis with only the forest ecosystem sites (n = 18) due to small sample size. The authors therefore really need to downplay this interaction or reanalyze with a larger dataset as it is currently not convincing. Note that forest loss and forest cover coefficients were also no longer positive with the +/- 95% CI (Supp. Fig. 4). This should be mentioned in the 'Sensitivity Analysis' section of the paper.

FG: With the increased sample size (major change #1), we are now able to run the same model selection process in the sensitivity analysis, and the results are largely consistent with the full dataset (see Supplementary Table 6 and Supplementary Fig.2).

5) Why not display the model-averaged coefficients for different taxa type (animal or plant)

since 'type' was a parameter in two of the competing models? It would also be valuable to split grassland versus forest dependent species (and maybe tropical) as I suspect that they responded differently to forest cover (%). Forest loss may also be correlated with other variables such as a gain in grassland cover and/or wetlands which could explain higher/faster movements for grassland and wetland dependent species. This additional piece would also probably help to explain the unexplained variance in the models at the species level.

FG: In the updated results, taxa type (plants vs. animals) had a weak effect at the aggregated level (cf. site level) and was not consistently significant in the candidate models listed in Table 1. For that reason, we decided not to show this result in the main text but as a Supplementary Figure (see the new Supplementary Fig. 1). We now discuss this weak effect in the main text but caution in reading too much into this effect (see lines 102-108 in the revised version). Regarding our analyses at the species level, taxa type (plants vs. animals) is not an important predictor (appeared in 1 out of 13 competing models) with a confidence interval straddling 0. Therefore, it is more appropriate to focus on the important and significant variables common among our candidate models. We understand the consideration for differentiating grassland and forest dependent species, but it is very difficult to apply a consistent classification criterion for the 2798 data points that cover a wide range of taxa. To address this concern, we ran the sensitivity analysis by restricting to forest systems only (assuming that species living in these areas are mostly forest dependent or forest specialists), and the consistent results further confirmed the robustness of the findings.

Minor comments:

Elsen & Tingley (2015) reference is missing from supplementary material.

FG: Added, thanks.

Pg 5, Line 84 – the variables of the best fitted model should be stated here.

FG: The variables (with detailed effect size and significance level) are now listed in Table 2.

Reviewer #2 (Remarks to the Author):

The authors looked at confounding effects of forest cover change on climate-driven range shifts across elevational gradients by compiling and re-analyzing published data. I think it is a much needed work even though studies focusing on climate-driven redistribution should have tried their best to exclude confounding factors, particularly habitat change, in the original design. In reality, it is challenging and not feasible for many studies. Thus, evaluating the interacting effects is a very nice attempt and provides insights into the widely reported biological responses under climate change.

After reading through the manuscript carefully, I think, generally, the authors analyzed the data appropriately given the question to address and data available so far. The discussion and conclusion followed logically and coherently. They also made it clear about the limitation of the study. However, I am not yet fully convinced by the interpretation of the results because of the constraints of multiple regressions, given the small sample size and some data treatment issues.

FG: We would like to thank referee #2 for her/his valuable comments. We have now significantly (almost doubled) increased our sample size (major change #1) and checked for data collinearity (major change #3). See detailed responses to specific comments below.

The authors have probably included most data relevant for addressing the research question. However, comparing to the number of predictors, the sample size was nevertheless small and the results may be sensitive to the data. There may exist multicollinearity or indirect effects among predictors. For example, higher forest cover (represented by "Cover" in the model) toward the tropics (baseline temperature, "T"), or, as the authors suggested in line 45-47, stronger deforestation ("Loss" in the model) toward the tropics. Indirect effects may occur for average temperature affects the level of forest cover and then buffers species responses. Both of them influence the robustness of multiple regressions and the indirect effect cannot be captured by current analysis setting. The authors could easily check the collinearities of variables. As for the indirect effect, however, the current sample size may still impede meaningful analysis.

FG: Our results are further strengthened by increasing the sample size to 43 (site-level) and 2798 (species-level) which is a very important improvement compared with the initial submission. We would like to thank again referee #2 for mentioning that issue as it pushed us to improve and consolidate our work. We also appreciate the thoughts on multicollinearity and have used the variance inflation factor (VIF) and applied a conservative cut-off threshold of 2 to address this potential problem. The variable "Gain" was thus removed in the species-level analysis due to its high VIF value, while all other variables have VIF values below the threshold and with low correlation coefficients (see the new Supplementary Tables 4-5). Therefore, we believe our results are robust to data collinearity.

The authors also conducted disaggregated analysis, which tried to look at species-specific responses and also increased sample size. However, because of the coarse spatial resolution of the climate data, a single temperature, identical to the baseline temperature for aggregated analysis, was applied to each species in the site of interest. The constraint made it difficult to test the authors' idea that lowland deforestation and/or forest expansion at highland may interact with climate to affect range shifts. The mismatch between species level response and their corresponding temperature also made the interpretation of baseline temperature

ambiguous. The elevational gradients could easily represent comparable temperature ranges along latitudinal gradients. Assigning an average temperature dramatically reduced the variation to site difference. It was thus not surprising that the disaggregated analysis derived generally similar results from aggregated analysis and unfortunately added limited information.

FG: Thanks for pointing out this issue. We have now replaced our temperature data using the newly published high resolution CHELSA dataset and hence are able to calculate fine-scale temperature for each 10-m elevation band as we did for the forest data (major change #2). We believe that by doing so, the aggregated and disaggregated analyses are more independent from each other. That we still find contributing effects from both habitat and climate variables in determining species range shifts in both analyses strongly supports our main argument for considering both drivers.

I had a closer look at the raw data and would like to know more details about the data treatment. As far as I understand, many studies looked at higher part of the elevational gradients to avoid impacts of lowland deforestation. For example, Jump et al. (2012) looked at plants at summit areas. Raxworthy et al. (2008) analyzed species distributed higher than about 1400 m.a.s.l. However, in Supplementary Figure 3, the authors seemed to include lower part of the gradient that was outside the focal extent of the original study. I did not check all of them. Please make it clear whether all the forest change profiles corresponded to the study gradients. This is where the authors argued not being considered in the previous studies but they might have done.

FG: We did calculate the forest and temperature data corresponding to the study gradients in all analyses. To make it less confusing we have now replotted the forest cover and change profiles by restricting to the focal elevation range of each study.

For the aggregated analysis, Chen et al. (2009) and Chen et al. (2011) both looked at moth assemblages in Mount Kinabalu and shouldn't be included twice. For disaggregated analysis, the number of species of each study included in the final analysis differed a lot from the aggregated analysis. The authors included 21 data points of Chen et al. (2011) from the original 130 species, 22 data points of Jump et al. (2012) from the original 24 species, 15 data points of Konvicka et al. (2003) from the original 119 species, 21 data points of Parolo and Rossi (2008) from the original 93 species...etc. For Walther et al. (2005), more species than originally reported were included (24 rather than 18). Please clarify if there were criteria to include species.

FG: Thanks for the suggestions, we have now excluded the Chen et al. (2009) study and only included the more updated 2011 dataset. In addition, we double checked our dataset to make sure that all information is correctly recorded (e.g. 24 data points of Jump et al 2012). We recorded the n of each site-level shift based on which the average shift rate was calculated. The smaller number of data points in the disaggregated analysis compared with the n in aggregated ones were often because that raw data was not available in the original study. In such cases, we tried to obtain as much species-level information as possible based on figures (e.g. Chen et al 2011) or tables reporting significant shifts (e.g. Konvicka et al 2003). [For the latter case the averaged site-level shift was adjusted by the shift proportion]. As for the increased n in Walther et al. (2005), the author focused on the shift on the same mountain summit ($n = 18$) while we believed that the broad shift across the Bernina area ($n = 24$) was more appropriate for our analysis.

More than half of the studies focused on plants. As the authors suggested, forest change implied habitat alteration and decreasing buffering effect on animal species under warming. However, for the plants, the gain and loss of forest may reflect direct population extirpation or increase. Shall we give broadly similar explanations to the underlying mechanisms?

FG: Thanks for pointing out the alternative mechanism, we have now included it in the manuscript. However, the main scope of this study is not to test specific mechanisms but to assess the confounding impacts of climate and land-use change. And the fact that we did not find a significant difference between taxa at the species level, although a weak but non-systematic effect was detected at the site level, suggests that the impacts of both factors are independent of taxa type (animals vs. plants).

Reviewer #3 (Remarks to the Author):

Guo and colleagues follow up on the landmark global study of Lenoir and Svenning (2015), who analysed 1030 elevational range shifts from 29 published studies. Here, the focus is on the impact of baseline forest cover and recent forest cover change (which they assessed using the Global Forest Watch database for post-2000 changes) on these documented range shifts, and on the interaction between these factors and baseline temperature and temperature change (estimated specifically for each study). They analysed the data in two ways—at the study level, and at the species level (while controlling for nested nature of the data).

I should say, first, that reading the Methods (when I finally got them) reassured me that authors had successfully steered clear the many statistical pitfalls that had come to mind as I read the Introduction and the Results. The analysis appears quite sound, and the statistical inferences (given the data—see below) appropriately conservative.

The authors found clear statistical evidence of a synergistic (amplifying) interaction between the forest factors and the climate factors. Few will be at all surprised by this finding, as it has been frequently predicted, but as far as I know this study is the first to offer rigorous evidence.

Less predicable (and thus potentially more important) is their finding that “forest loss positively interacts with baseline temperature conditions such that forest loss in warmer regions tends to accelerate species upslope movements.” Unless I am mistaken, this conclusion relies rather heavily on the only three tropical sites among the 29 sites: Hawaii (Volcanoes NP), Borneo (Mt. Kinabalu), and Madagascar. This modest site-level sample size is no fault of the authors—as far as I know, those three studies are the only tropical ones so far available that meet the Lenoir and Svenning’s (2015) selection criteria. The authors mention the small number of tropical sites as a limitation of their study (Line 186 ff), but I confess that their reassurance on this point left me scratching my head: “However, because our aim was to investigate the overlooked interaction between climate change and land-use change on species redistribution, instead of drawing a conclusive numerical model, we believe the data used here are suitable.”

FG: We would like to thank Robert K. Colwell for the very positive feedback on our initial work and for his insightful comments that helped us improve the quality of our work. As now explained in major change #1, we updated the Lenoir and Svenning (2015) dataset by adding a fairly large number of studies all published between 2014 to 2017 and have substantially increased the sample size, especially with 6 additional studies in the tropics (e.g. Asia, Central America and South America: see the updated version of Figure 2 which shows the improvement in spatial coverage with this updated dataset). Therefore, we believe that the interaction term that we still find between forest loss and baseline temperature is strong and coherent. We are very happy about this new and enhanced dataset that offers a consolidated overview of elevational range shifts thanks to the very recent reports from the tropics.

More troubling, perhaps, is that all three tropical sites are large islands, each famous in its own right for endemism and notorious for its lowland habitat destruction. How this fact might affect the results is not clear, but the potential bias should at least be discussed. For Hawaii and Madagascar (more isolated than Borneo), highland biotas (if memory serves) are believed to have been derived from lowland immigrants, which is surely not the case with most continental sites. (There is an excellent paper on the origins of the Mt. Kinabalu biota, but I can’t seem to put my hands on it.)

FG: Thanks for pointing out the tropical island bias. We have now included this point in our discussion. Note, however, that by including more continental sites in the tropics we have attempted to correct for this sampling bias, to some extent. In addition, we also think that the forest cover baseline in year 2000 could reflect historical habitat destruction across elevations whereas the recent shift recorded over the past few decades should be comparable across tropical and continental sites.

The authors note that “greater elevational distance to the highest mountain summit within the study areas” (Line 168) had an effect on the results, but they do not mention the Massenerhebung effect (see Grubb 1971, McCain 2005), which may well be relevant to this finding. (Mt. Kinabalu is, in fact, a classic case of this effect.)

FG: Thanks for the suggested references, we have now discussed and incorporated the Massenerhebung effect in the revised text.

Connectivity is (appropriately) mentioned as a critical issue in the Abstract, but not adequately addressed in the body of the paper. Does the Global Forest Watch data somehow take connectivity into account? Assessing connectivity quantitatively for each study area (the novel method of Forrero-Medina et al. 2010, cited by authors, might potentially be applied) may be too much to ask for this study, but it should be discussed.

FG: Indeed, quantitatively assessing forest connectivity would be ideal, yet unfortunately we don't have the actual distribution information (apart from elevation) that Forrero-Medina et al. 2010 relied on, hence we were not able to use the same method to assess connectivity. However, we believe that the forest cover percentage could represent forest connectivity across elevation, with denser forest better connected than sparse ones, albeit the pure effect of fragmentation per se after controlling for the confounding effect of area loss cannot be tested here due to a lack of data at the global extent.

Some smaller things:

Fig. 3 a and b. I like the clever way of plotting interactions, but the caption does not make clear that the two panels plot the results for different models, and why they differ. It took me way too long to figure it out from the text.

FG: Thanks, we have now clarified the figure caption to make it more intuitive. The figure also changed somewhat with the updated dataset.

Line 158. “The fact that warm-adapted tropical species tend to shift more rapidly upslope compared to those in cooler regions could be explained by seasonal temperature stability, i.e. ‘mountain passes are higher in the tropics’ (Janzen 1967; Bonebrake 2013).” I am baffled by this suggestion. Janzen’s point was precisely the opposite: in the tropics, elevational shifts (e.g. going up an over a mountain pass) are more challenging than at higher latitudes because species are more narrowly adapted to local (stable) temperature limits.

FG: We have clarified the statement to emphasize temperature stability and thermal niches (see lines 147-151 in the revised text).

Line 161. “Alternatively, it could also be partially driven by the shallow temperature gradient across latitudes in the tropics, which makes the other option, elevational shift towards mountain summits, more likely....” It might be more accurate to say, “Alternatively, it could also be partially driven by the shallow temperature gradient across latitudes in the tropics, which makes latitudinal shifts less likely than the other option, elevational shift towards mountain summits.”

FG: Changed accordingly. Many thanks for the suggestion.

Grubb, P. 1971. Interpretation of the ‘Massenerhebung’ effect on tropical mountains. *Nature* 229:44-45.

McCain, C. 2005. Elevational gradients in diversity of small mammals. *Ecology* 86:366-372.

Robert K. Colwell
University of Connecticut

Reviewers' comments:

Reviewer #1 (Remarks to the Author):

The authors have adequately addressed the limitation of a small sample size. However, the use of the terms 'montane' and 'strong interactions' are questionable throughout the manuscript. Not all studies included in this analysis are from montane environments (e.g., Zuckerberg et al. 2009 with breeding birds; Comte and Grenouillet 2013 with stream fish). In fact, Zuckerberg et al. (2009) reported poleward shifts but little change in their 129 bird species elevational boundaries... I would therefore recommend the authors to focus on 'elevational species redistribution' in the article and remove the term 'montane' from their abstract and conclusions.

I make my case for the overuse of the term 'strong interactions' in the examples listed here:

Pg 6 lines 92-94 "We found a consistent positive interaction effect between forest loss and temperature" – but not when the non-forest species were excluded (Supp. Fig. 2). This should be discussed in the discussion.

Pg 8 lines 126-127 "we found strong synergistic effects between baseline temperature and forest cover" – There not strong based on the confidence intervals. The word 'strong' should be reconsidered here.

Pg 9 lines 152-154 "generally positive effects of forest cover on the elevational shift rate" – Not based on the model average coefficients for the species level and when the outcomes were restricted to forest ecosystems only.

Pg 17 lines 342-344 "results were largely consistent" – Except the predictor scale (Loss) was no longer significant ($p > 0.05$) and based on Supp. Fig 2, Temperature and Forest Cover coefficient averages are no longer significant (see confidence intervals).

The conclusion of 'strong interactions' (Pg 11 lines 196-197) are therefore questionable based on these specific examples listed above.

Nicely done on the reanalysis, but why did you include taxa type as a fixed effect and not as a random intercept? You may not see a difference in the results, but it should at least be tested in your LMM random structure prescreening (pg 15, lines 309-311)... and may explain some of the unexplained variance at the species level.

I missed this during the first review, but where are the residual plots and goodness of fit tests? These are needed for the reviewers to determine the fit of the best fitting models.

Respectfully,
Chrystal Mantyka-Pringle

Reviewer #2 (Remarks to the Author):

The authors have put substantial effort to improve the robustness of the results by increasing sample size to 43 sites and 2798 species. They also applied fine-scale temperature to species specific analysis, making it more independent from site-specific analyses. From the revised results, the authors found generally positive effects of climate (base line temperature) and habitat (forest cover)

on range shifts. Importantly, they found synergistic effect between forest loss and baseline temperature on species redistribution in site level analysis, which constituted the main points of this work. Again, I agree it is a very thoughtful paper and here I have only a few questions left:

The temporal mismatch between the forest change (2000-2015) and the surveys of species range shifts (over several decades) was worrisome. The starting years of species survey were on average 44-year ahead the beginning of the forest database, and only the last 7 years (average across studies) were covered by the forest data, let alone some studies showed no overlap. The spatial consistency of the Global Forest Watch didn't ease the concern of methodological artefact. However, given data availability, there might be no better solution for this. I would suggest that the authors reveal and emphasize the temporal mismatch in greater detail, so that the readers may judge the results by themselves.

The positive correlation between forest loss and gain was counter-intuitive and raises similar concern about methodological artefact. Is there any particular reason for this phenomenon? If it was forest gain applied to the analysis, would it yield meaningful results to support current argument? I encourage providing different supporting evidence to strengthen current discussion.

The authors raised insightful questions about the opposite impacts of forest loss on species range shifts in warmer and colder regions. There may be post hoc explanations for the findings but in fact, plant extirpation or microclimatic changes following habitat disturbance were not restricted to warmer regions. And fundamentally, why wouldn't forest loss impede range shifts of warm-adapted species, at least in relative sense, comparing to cold adapted species? Given current findings, I guess the authors also need to call attention to the habitat-constrained climate tracking for temperate species.

A minor point about the baseline temperature in the site level analysis - it confounded with the extent of elevational gradient, not simply reflecting tropical or boreal sites. Many low latitude sites yielded similar baseline temperature to cooler regions. In this regard, species level analysis captured the temperature effects better.

For the sensitivity analysis, was the interaction term sensitive to the criteria of forest ecosystem? Current criteria, i.e. > 25% forest coverage seemed to be loose. Please also report the details of the interaction term.

Reviewer #3 (Remarks to the Author):

In this revision, Guo and colleagues have been exceptionally thorough in responding to reviewers' concerns and suggestions, and impressively enterprising in expanding the basis for their study to additional datasets, making possible additional and more rigorous statistical analyses.

In my earlier review, I had expressed concern about relying on just three tropical sites—all of them islands—to infer the important result that range shifts driven by forest loss are more rapid at low latitudes than in temperate and boreal regions. The authors have now discussed this issue and managed to find a few additional tropical sites.

The assumption made by the authors is that "baseline temperature" is a proxy for latitude, which it certainly can be. But shifts of high elevation tropical species could have a baseline temperature characteristic of boreal regions (3000 m at sea level on the equator corresponds to 50 degrees latitude at sea level). To my dismay, unless I am mistaken, latitude is not tabled for the study sites, although (of course) elevational range is recorded. I suggest adding latitude data to the Supplemental tables,

while making clear that "baseline temperature" is not a simple proxy for latitude.

Fig. 4. The X-axis labels are too small, and they should be defined in the caption.

Tables 1 and 2. Define the variables in the table caption.

There were many grammatical and usage problems in the main text. Being a compulsive editor, and having identified myself, I took the liberty of correcting them in the Word document, with Track Changes.

Robert K. Colwell
University of Connecticut

[Editorial Note: Due to journal policy, we are unable to publish the Reviewer #3 Marked Up Manuscript file as part of this Peer Review File.]

Reviewer #1 (Remarks to the Author):

The authors have adequately addressed the limitation of a small sample size. However, the use of the terms 'montane' and 'strong interactions' are questionable throughout the manuscript. Not all studies included in this analysis are from montane environments (e.g., Zuckerman et al. 2009 with breeding birds; Comte and Grenouillet 2013 with stream fish). In fact, Zuckerman et al. (2009) reported poleward shifts but little change in their 129 bird species elevational boundaries... I would therefore recommend the authors to focus on 'elevational species redistribution' in the article and remove the term 'montane' from their abstract and conclusions.

FG et al.: We agree that “elevational species redistribution” rather than “montane species redistribution” is more accurate and we changed the text of our revised manuscript accordingly, thanks for the suggestion.

I make my case for the overuse of the term 'strong interactions' in the examples listed here:

Pg 6 lines 92-94 “We found a consistent positive interaction effect between forest loss and temperature” – but not when the non-forest species were excluded (Supp. Fig. 2). This should be discussed in the discussion.

FG et al.: Here and throughout the revised manuscript, we emphasized the consistent interaction between forest loss and baseline temperature for all site-level analyses (including both full and forest-restricted datasets: see Supplementary Table.6) but not for the species-level analyses. The supplementary Fig.2 shows the results of the species-level analysis using the forest-restricted dataset, which, as consistent with the results from full dataset, highlighted contributing effects from both land-use and climate factors in species elevational range shifts. We are sorry for the potential confusion and we modified the text when necessary to clarify that when we mentioned the consistent interaction between forest loss and baseline temperature then it refers to the site-level analysis only.

Pg 8 lines 126-127 “we found strong synergistic effects between baseline temperature and forest cover” – There not strong based on the confidence intervals. The word 'strong' should be reconsidered here.

FG et al.: We replaced “strong” by “significant” as requested, thanks for pointing that out.

Pg 9 lines 152-154 “generally positive effects of forest cover on the elevational shift rate” – Not based on the model average coefficients for the species level and when the outcomes were restricted to forest ecosystems only.

FG et al.: For the species-level analysis using forest-restricted datasets, we found an additional (significant) synergistic interaction between “forest cover” and “baseline temperature” (See Supplementary Fig.2). Therefore, the individual effect of the variable “forest cover” in this case is dependent on the other variable (cf. baseline temperature), and it is not appropriate to interpret the individual effects of “forest cover” and “baseline temperature” regardless of the interaction term. Indeed, the coefficient estimate of the variable “forest cover” could be negative, but when the coefficient estimate of the interaction term between forest cover and baseline temperature is added to it, then the summed coefficient estimate may switch from negative to positive. That is why it is of utmost importance to avoid interpretation of the coefficient estimates of the individual effects of “forest cover” and “baseline temperature” when the interaction term is significant.

Pg 17 lines 342-344 “results were largely consistent” – Except the predictor scale (Loss) was no longer significant ($p > 0.05$) and based on Supp. Fig 2, Temperature and Forest Cover coefficient averages are no longer significant (see confidence intervals).

FG et al.: Similar to the point above, a significant interaction term between two variables suggests that both variables are important even though individual main effects are not significant themselves (e.g. Loss*T, T*Cover). This means that if the interaction term “Loss:T” is significant, then the individual effects of “Loss” and “T” cannot be removed from the model, even if non-significant. Same applies for “T*Cover”. This means that when an interaction term is significant between two variables, then the individual main effects of each variable cannot be isolated and should not be interpreted without accounting for the interaction term. Therefore, we found the results largely

consistent because the key contributing factors (Loss, T, Cover) remained the same, once the interaction terms are accounted for.

The conclusion of 'strong interactions' (Pg 11 lines 196-197) are therefore questionable based on these specific examples listed above.

FG et al.: Though we feel that the wording of “strong interactions” is justified given the above arguments, we understand the reviewer’s point and have changed this to “significant interactions” instead. This is a more conservative description of the results but does not detract from the overall important message and results of the study.

Nicely done on the reanalysis, but why did you include taxa type as a fixed effect and not as a random intercept? You may not see a difference in the results, but it should at least be tested in your LMM random structure prescreening (pg 15, lines 309-311)... and may explain some of the unexplained variance at the species level.

FG et al.: Thanks for your comment of the use of the variable “taxa type”. The reason why we used “taxa type” as a fixed factor and not a random term in our models is that we were specifically interested in testing the difference between the two studied groups (animals vs. plants). Besides, this was a comment from one referee on an earlier version of the manuscript. Moreover, as taxa type itself is a dichotomous rather than a multi-level factor variable (cf. only two levels), it is more appropriate to treat it as a fixed effect grouping variable rather than a random effect variable. Indeed, it has been argued that random terms in mixed-effects models should be restricted to factor variables with many levels (e.g. study site) and that factor variables with few levels should rather be treated as fixed effect terms. For all these reasons (ecological and statistical), we believe that it is more appropriate to treat “taxa type” as a fixed effect and not as a random effect. We recommend the very useful blog post as well as the responses to comments that are written by Brian McGill on this topic “Is it a fixed or random effect?”:

<https://dynamicecology.wordpress.com/2015/11/04/is-it-a-fixed-or-random-effect/>

I missed this during the first review, but where are the residual plots and goodness of fit tests? These are needed for the reviewers to determine the fit of the best fitting models.

FG et al.: Sorry about that. We did not display residual plots in previous versions. We now provide these plots for both the site-level and species-level analyses (see Supplementary Figs. 6-7).

Respectfully,

Chrystal Mantyka-Pringle

FG et al.: Many thanks for your time to review our work and provide insightful comments, which improved the quality of the manuscript.

Reviewer #2 (Remarks to the Author):

The authors have put substantial effort to improve the robustness of the results by increasing sample size to 43 sites and 2798 species. They also applied fine-scale temperature to species specific analysis, making it more independent from site-specific analyses. From the revised results, the authors found generally positive effects of climate (base line temperature) and habitat (forest cover) on range shifts. Importantly, they found synergistic effect between forest loss and baseline temperature on species redistribution in site level analysis, which constituted the main points of this work. Again, I agree it is a very thoughtful paper and here I have only a few questions left:

FG et al.: Many thanks for your very positive and encouraging words.

The temporal mismatch between the forest change (2000-2015) and the surveys of species range shifts (over several decades) was worrisome. The starting years of species survey were on average 44-year ahead the beginning of the forest database, and only the last 7 years (average across studies) were covered by the forest data, let alone some studies showed no overlap. The spatial consistency of the Global Forest Watch didn't ease the concern of methodological artefact. However, given data availability, there might be no better solution for this. I would suggest that the authors reveal and emphasize the temporal mismatch in greater detail, so that the readers may judge the results by themselves.

FG et al.: Indeed, there is no better data available but we appreciate the suggestion and we have now expanded our discussion on the limitation regarding the temporal mismatch. We assume that the baseline forest cover information (year 2000) could indirectly capture land-use history that the forest change data was not able to cover. We also hope the potential artefact could be partially eased by the large sample size of this study. Anyway, we now better emphasize this temporal mismatch, as requested, so that the readers may judge the results themselves.

The positive correlation between forest loss and gain was counter-intuitive and raises similar concern about methodological artefact. Is there any particular reason for this phenomenon? If it was forest gain applied to the analysis, would it yield meaningful results to support current argument? I encourage providing different supporting evidence to strengthen current discussion.

FG et al.: In the remotely sensed GFW dataset, forest gain was defined as a change from non-forest to forest state, and unlike the annually allocated forest loss information, forest gain percentage was calculated over the entire study period (2000-2012) due to the slow growing process. Natural growth of existing long-lived forest was not considered as forest gain (Hansen et al 2013). Therefore, forest loss and gain are often correlated in areas with temporary deforestation (e.g. logging, harvesting, fire, etc.) followed by replantation or natural regrowth. We believe that such temporary loss (along with forest gain) would still lead to drastic habitat disturbance and affect the associated biotic community. For that reason (cf. disturbance-related effects), we believe that forest loss is a more appropriate variable than forest gain to explain species elevational range shifts. To justify our choice (in addition to the fact that forest gain and forest loss are highly correlated), we now provide a sentence to explain why we used forest loss and not forest gain.

The authors raised insightful questions about the opposite impacts of forest loss on species range shifts in warmer and colder regions. There may be post hoc explanations for the findings but in fact, plant extirpation or microclimatic changes following habitat disturbance were not restricted to warmer regions. And fundamentally, why wouldn't forest loss impede range shifts of warm-adapted species, at least in relative sense, comparing to cold adapted species? Given current findings, I guess the authors also need to call attention to the habitat-constrained climate tracking for temperate species.

FG et al.: Thanks for this insightful comment. We agree that we cannot draw any conclusive mechanism underlying such interactions, based on correlative approaches, without practical experiments. For instance, the positive correlation between baseline temperature and forest cover could partly explain why forest loss impedes range shifts of cold-adapted species (cf. baseline forest cover is relatively low) more than warm-adapted species (cf. baseline forest cover is higher). Alternatively, it could also be due to the intrinsic physiological differences between these two groups of species. Indeed, cold-adapted species don't have to shift that much, compared with warm-adapted species, because of their relatively wider climatic niches. In other words, temperate species might be habitat-constrained to track climate, or simply adapt better so that they don't need to track changes. We appreciate the critical and objective thinking but we think what is more important is to emphasize the distinctive patterns rather than provide all hypotheses (of which

there are many possibilities). We have, in the revision, reworded to emphasize the uncertainty behind the potential underlying mechanisms.

A minor point about the baseline temperature in the site level analysis- it confounded with the extent of elevational gradient, not simply reflecting tropical or boreal sites. Many low latitude sites yielded similar baseline temperature to cooler regions. In this regard, species level analysis captured the temperature effects better.

FG et al.: Thanks for this insightful comment. We have also included the latitude information to distinguish temperature and location.

For the sensitivity analysis, was the interaction term sensitive to the criteria of forest ecosystem? Current criteria, i.e. > 25% forest coverage seemed to be loose. Please also report the details of the interaction term.

FG et al.: In the standardized definition of forest, the Food and Agriculture Organization of the United Nations (FAO) used a minimum threshold of 10% canopy cover. Therefore, we believe the 25% threshold we used here is conservative enough and allows for a sufficient sample size for model comparison. We have also applied a more conservative threshold of 30% and the results remained consistent. However, increasing the threshold to 40% and 50% yielded very low sample sizes not suitable for analysis.

Reviewer #3 (Remarks to the Author):

In this revision, Guo and colleagues have been exceptionally thorough in responding to reviewers' concerns and suggestions, and impressively enterprising in expanding the basis for their study to additional datasets, making possible additional and more rigorous statistical analyses.

FG et al.: Many thanks for your positive words. We really appreciate your feedback.

In my earlier review, I had expressed concern about relying on just three tropical sites—all of them islands—to infer the important result that range shifts driven by forest loss are more rapid at low latitudes than in temperate and boreal regions. The authors have now discussed this issue and managed to find a few additional tropical sites.

The assumption made by the authors is that “baseline temperature” is a proxy for latitude, which it certainly can be. But shifts of high elevation tropical species could have a baseline temperature characteristic of boreal regions (3000 m at sea level on the equator corresponds to 50 degrees latitude at sea level). To my dismay, unless I am mistaken, latitude is not tabled for the study sites, although (of course) elevational range is recorded. I suggest adding latitude data to the Supplemental tables, while making clear that “baseline temperature” is not a simple proxy for latitude.

FG et al.: Many thanks for this suggestion, we have now tabulated the latitude information for each study site and further emphasised the effects of temperature itself while being clear that baseline temperature is not a simple proxy for latitude.

Fig. 4. The X-axis labels are too small, and they should be defined in the caption.

FG et al.: Changed accordingly, thanks.

Tables 1 and 2. Define the variables in the table caption.

FG et al.: Done.

There were many grammatical and usage problems in the main text. Being a compulsive editor, and having identified myself, I took the liberty of correcting them in the Word document, with Track Changes.

FG et al.: We are very grateful for all the constructive feedbacks and corrections. Many thanks for your edits directly into the text. We really appreciate your help to improve the quality of the text.

REVIEWERS' COMMENTS:

Reviewer #2 (Remarks to the Author):

The points raised in the previous round of review have been satisfactorily addressed.

REVIEWERS' COMMENTS:

Reviewer #2 (Remarks to the Author):

The points raised in the previous round of review have been satisfactorily addressed.

FG et al.: We are grateful for your positive feedback. Thanks for helping to improve the quality of our work.